# Cognitive and Cultural Factors That Affect General Vaccination and COVID-19 Vaccination Attitudes

**DOI:** 10.3390/vaccines11010094

**Published:** 2022-12-30

**Authors:** Alla Keselman, Catherine Arnott Smith, Amanda J. Wilson, Gondy Leroy, David R. Kaufman

**Affiliations:** 1National Library of Medicine, National Institutes of Health, Bethesda, MD 20894, USA; 2Information School, University of Wisconsin-Madison, Madison, WI 53706, USA; 3Management Information Systems, University of Arizona, Tucson, AZ 85721, USA; 4Medical Informatics Program, SUNY Downstate Health Sciences University, Brooklyn, NY 11203, USA

**Keywords:** COVID-19, information literacy, misinformation, science literacy, trust, vaccination hesitancy

## Abstract

The development of COVID-19 vaccines is a major scientific accomplishment that has armed communities worldwide with powerful epidemic control tools. Yet, COVID-19 vaccination efforts in the US have been marred by persistent vaccine hesitancy. We used survey methodology to explore the impact of different cognitive and cultural factors on the public’s general vaccination attitudes, attitudes towards COVID-19 vaccines, and COVID-19 vaccination status. The factors include information literacy, science literacy, attitudes towards science, interpersonal trust, public health trust, political ideology, and religiosity. The analysis suggests that attitudes towards vaccination are influenced by a multitude of factors that operate in a complex manner. General vaccination attitude was most affected by attitudes towards science and public health trust and to a lesser degree by information literacy, science literacy, and religiosity. Attitudes towards COVID-19 vaccines were most affected by public health trust and to a lesser extent by general trust, ideology and attitudes towards science. Vaccination status was most influenced by public health trust. Possible mediating effects of correlated variables in the model need to be further explored. The study underscores the importance of understanding the relationship between public health trust, literacies, and sociocultural factors.

## 1. Introduction

Many factors define the public’s acceptance or rejection of public health advice, particularly in the context of COVID-19 vaccination. This work investigates several of these factors. At the time writing, the COVID-19 pandemic has been ongoing for almost three years, claiming about 1,101,880 lives in the US alone [1]. In December 2020, the FDA issued an Emergency Use Authorization (EUA) making the first COVID-19 vaccine currently approved in the US available to the public [2]. The development of vaccines, endorsed as safe and effective by the Centers for Disease Control and Prevention [CDC] [3], has been a major public health victory in the fight against the new virus. 

The story of COVID-19 vaccination has been tarnished with persistent vaccine hesitancy becoming part of the narrative and public health legacy. Many Americans have expressed doubts about the safety and effectiveness of vaccines, despite the data and encouragement provided by the CDC, health professionals, and federal and local governments. The phenomenon of COVID-19 vaccination hesitancy and misinformation spread are not limited to the US [4,5,6,7,8]. While the level of vaccine hesitancy was striking in the face of a pandemic of a novel virus, it illustrates the established phenomenon of the ubiquity and power of health information running counter to what is produced by the biomedical establishment. 

Calleja et al. [9] proposed a public health research agenda emanating from the first WHO Infodemiology Conference. These authors note the paucity of research into the relationship between offline and online behaviors and suggest that more work is needed to understand “indicators that predict behaviors or serve as proxies for specific behaviors.” The present work attempts to fill these gaps, first, by exploring the connections between vaccination attitudes and multiple cognitive and cultural factors, and second, by uncovering the differences between general vaccination attitudes and COVID-19-specific attitudes with respect to information behaviors. These cognitive and cultural factors are not usually considered in combination, and some of them (e.g., science literacy defined as knowledge about science) have not previously been considered in studies of vaccination hesitancy.

### 1.1. Factors That Affect Vaccination

“Vaccine hesitancy” has been defined by the World Health Organization (WHO) as “the delay in acceptance or refusal of vaccines despite the availability of vaccination services” [10]. Factors contributing to vaccine hesitancy identified in systematic and rapid reviews since 2014 include gender differences [11]; access to information resources; concerns about side effects; a lack of awareness; physician advice; costs and lack of availability [12]; younger age, lower-income status, lower education level, and ethnic minority group membership [11]; absence of provider’s advice and recommendation to vaccinate [13]; inefficiency of government vaccination initiatives [7]; and the WHO’s triumvirate of known weaknesses: complacency, (low) confidence, and inconvenience (in accessing vaccines) [14]. In reviews of COVID-19 vaccine hesitancy studies across the world, recurring themes are lower perception of risk and severity, concerns about vaccine safety, and lower trust in governmental and societal institutions [15,16]. The WHO Vaccine Hesitancy Matrix considers characteristics influencing vaccine hesitancy in three broad categories: *contextual* (historic, sociocultural, etc.)*, individual/group* (e.g., personal attitudes and social effects)*,* and *vaccine-specific* (e.g., mode of administration and costs) [10]. In this study, we approach vaccination from a cognitive–cultural perspective, focusing on individual factors. Cognitive studies explain that individuals selectively process information, giving more weight to arguments that are coextensive with their biases [17]. These cognitive factors impact on opinion-formation and decision-making and interplay with social and cultural factors. Individuals become trapped in homogenous information echo chambers with those who have similar interests and beliefs [18]. Kahan’s [19] identity protective mechanism (IPC) construct further explains the individual tendency to accept or reject beliefs associated with particular groups, such as political parties. The IPC can also serve as a filter for information sources and institutions that are deemed trustworthy or not. Personal affiliations hold more sway and may cause an individual to bypass rational thought, deliberative processes, and factually accurate beliefs. 

We investigate two cognitive factors, science literacy, including attitudes toward science, and information literacy. These factors are sets of skills and knowledge essential to navigating information and psychologically predisposing individuals to certain beliefs about vaccination safety and efficacy [20]. The cultural factors considered in this study pertain to political ideology/party affiliation and religiosity. We also focus on several kinds of trust, in which the cognitive and the sociocultural dimensions are closely intertwined. While this study was conducted from a US perspective, the literature suggests that similar factors are influential worldwide.

#### 1.1.1. Science Literacy

The way in which individuals relate to the scientific enterprise in their life and society is affected by their science-related competencies, described as science literacy. Studies of science literacy’s impact on vaccination attitudes typically measure science knowledge [21,22] or science education [23]. From a theory-driven perspective, science literacy is a multi-dimensional concept that involves the following [24,25]:a.Knowledge of scienceb.Knowledge about sciencec.Attitudes towards science

Competencies in these three dimensions may have different effects on daily life [26]. Promoters of science education often express the view that knowledge of science is important and should be taught to non-scientists. General science knowledge correlates with more positive attitudes towards vaccination in general [21], and COVID-19 vaccination specifically [22]. Yet, overall, capturing the impact of science knowledge on daily life in research studies has been difficult [27]. This does not necessarily mean that science knowledge has no effect on real life decisions. However, its impact is probably smaller than that of other factors; it may work in specific types of situations and in indirect ways.

Knowledge about science refers to understanding how science works as a process and an enterprise [25]. It includes understanding of methodologies for controlling bias, such as randomization and the control of variables, as well as institutional practices, such as science funding and peer review. While members of the public do not conduct scientific research, understanding the process is likely to enable them to approach science and health information critically. Similar to the knowledge of science, the impact of knowledge about science on daily life has not been widely researched. Keselman et al. [28] showed that greater knowledge about science reduces individuals’ readiness to share non-evidence-based health information on the Internet. However, knowledge about science is only taught in K-12 schools in a perfunctory way and in simplistic contexts [29].

Attitudes towards science encompass individuals’ beliefs about the advantages and disadvantages of science as a way of generating knowledge about the world, science’s impact on the world, and the motives of people conducting science [30]. The literature shows that while most Americans have positive attitudes towards science, a significant minority have concerns [30]. These concerns are differentially distributed in different population groups, discussed below in the Trust section.

#### 1.1.2. Information Literacy

Another individual factor that may influence health beliefs is information literacy, or “a set of aptitudes to locate, handle, evaluate, and use information efficiently for a wide variety of purposes” [31]. Numerous researchers have investigated the connection between *health* literacy and vaccination attitudes [32] and *media* literacy in relation to information valence [33]. However, *information* literacy has received much less attention [34]. 

Chan et al.’s [12] systematic review found that both *mis*information and a *lack* of information influenced patients’ vaccination hesitancy. While multiple literacies-“health, information, digital, and media”-are implicated in infodemics, “there is limited research on how exposure to information or misinformation affects behavior because behavioral processes can be quite complex” [9]. In one study, low and moderate levels of information literacy—but not high levels—significantly affected participants’ responses to a critical thinking recommendation in which they were asked to “stop and think” about the news content to which they were exposed [35]. The ability to search, evaluate, and verify COVID-19 vaccination information has been correlated with reduced vaccine hesitancy and increased vaccine uptake [8,36].

Information literacy competency may be influenced by many factors, including science literacy. In Keselman et al. [28], information literacy correlated with knowledge about science. It is possible that knowledge about science influenced information literacy by leading the individual to be more critical in selecting information sources, which, in turn, reduced the willingness to share unreliable information.

#### 1.1.3. Trust, Ideology, and Culture

Modern biomedical knowledge is extremely complex; thus, our acceptance of it ultimately must be grounded in trust in experts. Predispositions to trust or mistrust are determined by cultural experience of individuals and groups. In general, people differ in how much they trust others; interpersonal trust is higher in well-functioning communities and democratic societies [37]. Across the world, trust in the government, the healthcare system, and civil society correlates with COVID-19 vaccine acceptance and uptake [38,39,40]. In the US, studies have revealed that distrust of pharmaceutical companies, with them being perceived as being motivated by profit [41] and, overall, trust in the U.S. federal government have been waning since the Vietnam War era [42]. Yet, declining trust in the “government in Washington” is not correlated with levels of trust in people “running the institution” of science—even though the federal government is the major science funder in the US [43]. 

Individuals’ trust in institutions developing and endorsing COVID-19 vaccines is affected by political ideology and religiosity. Milligan et al. [44] found that less religious participants were more likely to accept a vaccine, as did Orlandi, Febo, and Perdichizzi [45] in a study of 22 European countries. Political and religious conservatives are less trusting of scientists, doubting the honesty and good intentions of the people running the scientific enterprise [30]. Conservative mistrust of scientific outputs varies across topics, with ideological distrust greater for topics marked by political partisan division, for example, climate change and evolution [46]. Unfortunately, COVID-19 vaccines have become a partisan topic in the US, and ideological conservatives are both more likely to have concerns about the safety and effectiveness of COVID-19 vaccines and are less likely to be vaccinated [47]. In a US survey measuring several social determinants of health, vaccine acceptance was lower among Whites with republican ideology [48]. This trend is not confined to the US. In a European survey, Debus and Tosun [49] found ideological extremism strongly related to vaccination skepticism. 

Among underserved groups, trust in government institutions is reduced by a history of racism and discrimination [41,50]. Members of these communities may also experience medical mistrust, the consequence of inequities in the quality of healthcare they receive [41,51,52]. These individuals may view members of the healthcare establishment—researchers, practitioners, and policy makers—as hostile outsiders [30]. These beliefs may influence health decisions [53]. For example, Evans and Gusmano [54] showed that vaccine hesitancy among Black mothers is influenced not only by medical mistrust, but also a mistrust of science and the government.

#### 1.1.4. Specific Objectives

The specific objectives of this study include investigating the impact of information literacy, science literacy, attitudes towards science, interpersonal trust, public health trust, political ideology and affiliation, and religiosity on three dependent variables: (1) general attitude towards vaccination, (2) attitude towards COVID-19 vaccination, and (3) COVID-19 vaccination status. We were interested in vaccination status as a behavioral variable of public health importance, related to our attitudinal variables. We also aimed to provide descriptive analysis of the distribution of key independent variables in the sample.

## 2. Materials and Methods

### 2.1. Participants

A convenience sample of 140 participants was recruited via the Amazon Mechanical Turk (MTurk) crowdsourcing platform. The sample size was determined by power assessment of our previous studies of health information behavior with a comparable number of variables [28], as well as by time and budget constraints. MTurk was chosen because it has a uniquely large U.S. worker base, estimated to be 75% of its worker membership [55], which is our target group. Moreover, the MTurk platform is well-known in the research community, making this sampling approach transparent to readers, and the authors have ample experience using the platform. The literature suggests that survey responses collected via MTurk are reliable and comparable to those collected via more traditional methods [56]. The inclusion criteria were being 18 or older, proficient in English, comfortable using the Internet, and residing in the US. We also required participants to have completed at least 50 MTurk tasks with 95% of those judged acceptable by task requestors. The MTurk counter was set to 150 participants; ten responses were dropped because they did not pass the data check inclusion criteria. Each participant received a $15 Amazon gift card. The participants’ demographic characteristics; religion and political affiliation; and vaccination status in August 2021 are described in Table 1, Table 2 and Table 3. 

### 2.2. Variables

The analysis included three dependent and nine independent variables (Table 4). 

### 2.3. Measures

All instruments are available as Appendix A. This section describes the measures underlying each of the variables.

#### 2.3.1. General Attitude towards Vaccination

This measure, adopted from previous research [58,59,60], was a composite of agreement scores with the following four statements:People who do not get vaccinated risk becoming very sickVaccines are effective in preventing diseasesI can get sick from vaccinesI am concerned that there may be something I do not know about in vaccines

Participants reacted to each statement by choosing one of five Likert scale response options ranging from “strongly disagree” to “strongly agree.”

#### 2.3.2. Attitude towards COVID-19 Vaccines

This measure was assessed via the participants’ agreement with the CDC statement that “COVID-19 vaccines are safe and very effective” [3]. Participants chose one of five Likert scale response options ranging from “strongly disagree” to “strongly agree.”

#### 2.3.3. Vaccination Status

This four-point scale measure reflected the participants’ choice of one of the following options:Received one or both dosesPlan to get vaccinated soonConsidering getting vaccinated in the future, but do not feel ready yetDo not want the vaccine

#### 2.3.4. Importance of Religion in Life

This measure reflected the participants’ rating of the importance of religion in their life on a five-point Likert scale, ranging from not at all important to extremely important.

#### 2.3.5. Frequency of Attending Religious Services

This measure asked participants to describe their frequency of attending religious services as “at least once a week,” “once or twice a month/a few times a year,” or “seldom/never.”

#### 2.3.6. Political Party Affiliation

This measure asked the participants to define their affiliation by choosing between “Republican or leaning Republican,” “No leaning/Independent,” “Democrat or leaning Democrat,” or “Other,” following US census categories.

#### 2.3.7. Political Ideology

This measure asked the participants to describe their political ideology as “very conservative,” “conservative,” “moderate,” “liberal,” “very liberal,” or “not sure,” following US census categories.

#### 2.3.8. Information Literacy

Information literacy was assessed using a six-question survey developed by the authors based on several existing instruments [61,62], and it was deemed to have acceptable levels of internal consistency in this team’s previous research (see [28]). The questions focus on assessing the participants’ awareness of markers of information authoritativeness and objectivity. Figure 1 presents a sample question item. 

#### 2.3.9. Science Literacy

Science literacy was assessed using a 12-question survey developed by the authors on the basis of test of scientific literacy skills (TOSLS) [62] and was deemed to have acceptable level of internal consistency in this team’s previous research [28]. The survey focused on two science-related skills, relevant to dealing with online health information, *identifying a valid scientific argument* and *understand elements of research design and how they impact scientific findings or conclusions*. Figure 2 and Figure 3 contain examples of science literacy test items. 

#### 2.3.10. Attitudes towards Science

The participants rated their agreement with the following five statements by selecting from the “agree or mostly agree,” “neither agree nor disagree,” or “mostly disagree or disagree” options.

Science is the most reliable way of learning about the natural worldAdvances that are made in science are relevant to me and my communityScientific research with human subjects protects people who participate in itScience does more good than harm in the worldI believe in science, but I do not trust scientists (e.g., because they may have other agendas)

The questions were developed on the basis of a literature review of research into attitudes towards science and trust in science [30]. Each response received the score of 0, 1, or 2 corresponding to the three response options, based on the expressed level of positive predisposition towards science. 

#### 2.3.11. Interpersonal Trust

The participants answered one question, commonly used as an interpersonal trust measure [63], “Generally speaking, would you say that most people can be trusted or that you cannot be too careful in dealing with people?” Possible answers ranged from one (“you cannot be too careful”) to five (“most people can be trusted”).

#### 2.3.12. Public Health Trust

The participants completed a survey about their trust in organizations and agents that typically represent the public health perspective:Centers for Disease Control and Prevention, CDCNational Institutes of Health, NIHYour primary doctor or healthcare providerA major university that conducts biomedical researchA national health association, such as the American Diabetes AssociationFood and Drug Administration, FDA

The answer options ranged from one (“do not trust at all”) to five (“trust completely”); for the analysis, the scores were converted to the 0–4 scale. The survey was an expanded version of the measure developed by this team based on a literature review of trust in health and medicine [64] and was deemed to have acceptable levels of internal consistency [28]. 

### 2.4. Data Collection and Preparation

We used Qualtrics XM software to collect the survey data. The survey link was delivered via the MTurk platform. The collection consisted of 150 responses. Of these, we removed those that were completed in under 6 minutes (based on pilot assessment of approximately 9 minutes needed for fast but attentive completion). The survey contained two attention questions. We also removed responses that failed both attention questions. Those that failed only one of the attention questions were reviewed for coherence (e.g., absence of contradictory answers) and found acceptable. At the end, 140 responses were included in the analysis.

### 2.5. Data Analysis

As the variables used in the quantitative analysis comprised responses to multiple-choice questions, data coding was not required. Each variable score was computed as a count of correct responses. Analysis was performed via SPSS (IBM) software.

## 3. Results

This section presents the findings of inferential statistical analysis followed by descriptive analysis of significant variables.

### 3.1. Single Predictor Statistical Models

As expected, based on prior research, independent variables were highly correlated. Modeling the approach on our previous work [28], we chose to first compare effect sizes of single predictor linear regression models. Individual effect sizes are important because, while the variables are likely to exert mutual multidirectional influences, interventions are often only able to focus on one variable or a small subset of variables. To apply Bonferroni correction for multiple hypotheses, the significance of individual models was assessed at *p* < 0.006. As all of the dependent variables were continuous (numerical or ordinal), linear regression was applied (Table 5 and Table 6).

### 3.2. Multiple Predictor Statistical Models

For each of the dependent variables, we performed linear regressions with all nine variables in the model. Multicollinearity diagnostic tests did not raise concerns about violation of assumptions for regression (all VIFs ≤ 3.55).

For the general vaccination attitude, the multivariate model was statistically significant F (9, 131) = 13.09, *p* < 0.001, r = 0.69 and explained 48% of the variance in the attitude towards COVID-19 vaccination. The effect size of this model (r > 0.5) is considered “large” [65]. The significance was attributable to two independent variables: attitudes towards science and public health trust. General trust was approaching significance. 

For the COVID-19 vaccines attitudes, the multivariate model was statistically significant F (9, 131) = 18.76, *p* < 0.001, r = 0.75 and explained 57% of the variance in the attitude towards COVID-19 vaccination. The significance was attributable to three independent variables: public health trust, general trust, and party affiliation. Details of the significant statistical values for both types of vaccination attitudes are presented in Table 7.

For COVID-19 vaccination status, the multivariate model was statistically significant F (9, 131) = 5.71, *p* < 0.001, r = 0.52 and explained 28% of the variance in the vaccination status. The size effect is considered “large” [65]. The significance was attributable to public health trust, t (1, 139) = 4.56, *p* < 0.001. 

### 3.3. An In-Depth Look at Key Independent Variables

In this section we present descriptive analysis of the distribution of key independent variables in our sample, as well as the correlations among them, with the expectation that they may provide insights for developing targeted interventions.

#### 3.3.1. Public Health Trust

Public health trust as a predictor had the highest effect size for two dependent variables, Covid vaccines attitudes and vaccination status, and was the second highest predictor for general vaccination attitude. Overall, public health trust in our sample was relatively high, M = 17.76, SD = 4.87 out of 25. Participants’ trust in the six authority institutions comprising this variable are presented in Table 8.

Of the other independent variables, public health trust was moderately strongly correlated (0.5 ≤ r < 0.7) with positive attitudes towards science (r = 0.56) and weakly correlated (0.3 ≤ r < 0.5) with information literacy (r = 0.39), ideology (r = 0.34), and generalized trust (r = 0.31). It was also weakly negatively correlated with importance of religion (r = −0.31).

#### 3.3.2. Attitudes towards Science

Attitude towards science had a high positive effect on general vaccination attitude, a moderate positive effect on Covid vaccines attitude, and a small positive effect on vaccination status. Overall, attitude towards science was rather positive in our sample, M = 7.49, SD = 2.23 out of 10. We were also interested in characteristics that co-occurred with positive attitudes towards science. In our sample, attitude towards science was moderately strongly correlated (0.5 ≤ r < 0.7) with information literacy (r = 0.58) and public health trust (r = 0.56) and weakly correlated (0.3 ≤ r < 0.5) with science literacy (r = 0.39). It was also weakly negatively correlated with importance of religion (r = −0.47). The relationship between attitudes towards science and ideology, frequently noted in the literature, is represented in Table 9. To represent that relationship, we divided attitudes towards science into negative (scores 0–3), cautious (4–7), and positive (scores 8–10). However, the correlation is very low (r = 0.19). 

#### 3.3.3. Information Literacy and Science Literacy

The average information literacy score was M = 3.96, SD = 2.13 out of 6 possible (Figure 4). The average science literacy score was M = 7.91, SD = 2.68 out of 12 possible (Figure 5). The two variables had the highest correlation of all the pairs of independent variables in the study, r = 0.77. 

In addition, information literacy was moderately strongly correlated (0.5 ≤ |r| < 0.7) with importance of religion (negatively, r = −0.62) and frequency of attending services (negatively, r = −0.61) and trust in science (positively, r = 0.58) and positively weakly correlated (0.3 ≤ r < 0.5) with public health trust (r = 0.39).

Science literacy was moderately negatively strongly correlated with religious attendance (r = −0.63) and importance of religion (r = −0.58) and weakly correlated with trust in science (positively, r = 0.39) and generalized trust (negatively, r = −0.30).

### 3.4. Concerns about COVID-19 Vaccination

While 73 (52.14%) of our participants “strongly agreed” and 46 (32.86%) “somewhat agreed” with the CDC that “COVID-19 vaccines are safe and very effective,” 8(5.71%) “neither agreed nor disagreed”, 7 (5%) “somewhat disagreed”, and 6 (4.29%) “strongly disagreed.” In this section, we consider specific concerns expressed about vaccines and agents or events that could have changed attitudes. 

Table 10 presents the participants’ response to “Which of the following is among your reasons for doubting getting the vaccine?” While 73 (52.14%) participants checked “no concerns,” 67 expressed one or more concerns, with the concern about vaccines’ safety being the most prevalent.

In narrative explanations of their safety concerns, the participants mentioned worries about the short-term and, more frequently, long-tern effects of vaccines, expressed concern about the fast vaccine development and approval process, not enough data for making decisions, and difficulty deciding which information to trust. Participants who expressed doubts about vaccines’ effectiveness pointed to cases of vaccinated individuals contracting COVID-19.

### 3.5. Information Sources and Social Influences

Table 11 lists information sources where participants reported obtaining their information about COVID-19 vaccines in the order of frequency (“Where do you obtain your information about COVID-19 vaccines?”). The data illustrate that, while newspapers and the CDC were the two most used information sources, more than half of the participants used social media as a source of COVID-19 vaccination information and 20% used web or mobile forums.

Table 12 presents different information agents in the order of perceived trustworthiness (“How much do you trust the following sources to help you locate reliable information about COVID-19 vaccination?”). The participants rated their degree of trust in these agents on a scale from one (“do not trust at all”) to five (“trust completely”). Comparing the data in Table 11 and Table 12 suggests that there are discrepancies between the most used and most trusted sources. For example, while participants felt that primary doctors or health care providers were the most trustworthy source of COVID-19 vaccination information, only a minority actually obtained their information from that source. In contrast, while newspapers constituted the most commonly used information source, journalists were trusted the least.

Participants also answered how various situations would change their interest in receiving the vaccine. Participants’ interest in the vaccine would be most positively increased (92 participants, or 65.71%) by learning that several family members and close friends had been vaccinated without experiencing side effects. This was followed by reading a CDC statement that “COVID-19 vaccines are safe and very effective” (89 participants, or 63.57%), and then by a doctor’s or healthcare provider’s suggestion to get vaccinated and reading “a detailed CDC explanation of how vaccines work” (85 participants, or 60.71% for each). Faith leader and celebrity endorsements were among the least effective interest-raising factors in our sample. The responses demonstrate the importance of the behavior and experiences of participants’ social peers, such as friends and family members.

## 4. Discussion and Conclusions

The public’s attitudes towards vaccination are influenced by a multitude of complex factors. In our sample, attitudes towards general vaccination were most affected by attitudes towards science and public health trust and to a lesser degree by information literacy, science literacy, and religiosity. It is likely that information literacy and science literacy influenced public health trust and positive attitudes towards science, mediating their impact on positive attitude towards vaccination. The impact of these two literacies on real-life health-related decisions is consistent with the findings of our previous work [28]. The impact of the two political ideology variables on general vaccination attitudes was significant but small and also, likely, indirect. The interplay of religiosity, science and information literacies, and attitudes toward science as a pathway to affecting vaccination attitudes merits further attention. 

Our study suggests that the factors shaping attitudes towards COVID-19 vaccination may be somewhat different from those implicated in general attitudes towards vaccination. This is consistent with other research finding that people think about politically controversial scientific topics differently than non-controversial ones [66]. In this study, public health trust was the strongest predictor of positive attitude towards COVID-19 vaccination. Attitude towards science was significant in a single-predictor model, suggesting that public health trust mediates the effect of attitude towards science. While the effect of party affiliation was small, it appeared to have a direct impact on attitude towards COVID-19 vaccines. It is noteworthy that science literacy, which had a moderate effect on general attitude towards vaccines, had no significant effect on attitudes towards COVID-19 vaccination specifically. Although our finding on the lack of impact of science literacy on COVID-19 vaccination attitude is discrepant with Siani, Carter, and Moulton [23] and Motoki, Saito, and Takano [22], those studies conceptualized science literacy as science education or knowledge of science, rather than knowledge about science. It is also noteworthy that party affiliation had a greater impact on attitudes towards COVID-19 vaccines than on vaccination in general, while religiosity, which had a moderate effect on general vaccination attitude, was not associated with attitudes towards COVID-19 vaccines. The present finding of the lack of association between religiosity and attitudes toward COVID-19 vaccination constitutes an intriguing discrepancy from the findings by Milligan et al. [44] and Orlandi, Febo, and Perdichizzi [45].

Several factors emerging as relevant to vaccination attitudes in our study converge with other investigations of predictors of COVID-19 vaccination attitudes. For example, our finding of the significance of public health trust is consistent with the findings of Chen, Lee, and Lin [38], Cvjetkovic et al. [39], and De Freitas, Basdeo, and Wang [40] about the positive impact of trust in the government, healthcare, and civil society on openness to COVID-19 vaccination. The importance of party affiliation is consistent with Khubchandani et al. [67] and Agarwal et al. [48] and the importance of information literacy with Engelbrecht, Kigozi, and Heunis [36] and Takahashi et al. [8]. Religiosity, which has been shown to affect COVID-19 vaccination attitudes [44,45], was relevant to general, but not to COVID-19-specific vaccination attitudes in our study. 

Our three dependent variables all have the same most impactful predictors, which are public health trust and attitudes towards science. The convergence of these predictors validates their significance, making them unlikely to be spurious. At the same time, their greater impact on vaccination attitudes, as opposed to actual vaccination status, illustrates the gulf between attitudes and behavior. While the design and the sample size of our study do not allow us to create a clear mediation model that would explain the pathway of impact of the multiple mutually influential independent variables on the dependent ones, a follow-up study that includes a large dataset collected from a more nationally representative sample would be amenable to structural equation modeling. 

As our sample, recruited via the MTurk platform, was small and not representative of the US population, our data do not permit us to make a conclusion about the state of information literacy and science literacy across the population. For example, according to the 2020 US census data, 37.5% of the population over 25 years of age had a bachelor’s degree or higher, compared to 66% in our sample [68]. Our sample also overrepresented Democrats vs. Republicans and Independents and liberals vs. conservatives. This sample, however, does give us a glimpse into COVID-19 vaccine safety-related concerns in a group that is more open to receiving them than the general population [67]: 26% doubt the safety of the vaccines, largely citing their novelty, rapid development, and insufficient research data backing their safety. We also glimpse the importance of social factors in opinion formation. While they trust the CDC and health professionals, participants are most likely to consider opinion change when seeing members of their close circles get vaccinated without negative consequences. Future surveys should include other online platforms, e.g., Clickworker (https://www.clickworker.com/) (with 4.5 million workers, 46% in the U.S.) and should focus on analyzing how the representative demographics of the participants recruited via different platforms affect survey responses.

### 4.1. Bringing about Attitude Change

Convincing the public that COVID-19 vaccines are safe and effective has been a major challenge for public health agents worldwide. This study underscores the importance of building public trust in scientific and public health establishments to achieve this goal. More research is important for developing strategies for gaining the trust of different population groups. For example, in our sample a minority of participants said that their attitudes towards COVID-19 vaccination could be influenced by faith leaders, community groups, and celebrities. These numbers, however, may be much higher in different communities [69]. In thinking about ways to build trust in public health organizations and positive attitudes towards science, we also need to consider both short-term and long-term efforts. During a pandemic, it is important to alleviate vaccination safety concerns fast. While both information literacy and science literacy may underlie attitudes towards science, and both literacies may be important, science literacy is generally the outcome of years of science education and exposure to science practice. Thus, science literacy change is not a good candidate for targeting with a quick time-sensitive initiative. At the same time, sustained long-term work with K-12 schools and informal science education settings, such as via social vaccine campaigns [70], is very important.

### 4.2. Directions for Future Research

It is important to gain an in-depth understanding of the independent variables used in this study, as well as build the interactive model of their impact on information behaviors and health beliefs. As such a model is likely to be very complex and context-dependent, it is practical to focus on subsets of variables and draw on methodologies of different fields. For example, the relationship between knowledge of science, knowledge about science, and attitudes towards science has received increased attention in science education (for a review, see [26]). Other studies have focused on the role of ideology and religious beliefs on attitudes towards science (for a review, see [30]). Multiple research approaches could provide different pieces of the puzzle. For example, large quantitative data sets amenable to structural equation modeling could help identify complex relationships among multiple variables. The sample size employed in this study is based on heuristics, precedent, and pragmatic considerations. This choice was made because it would have been impossible to complete a power calculation given the relatively scant literature on the investigated issues and the absence of effect-size measures. Our sample size is in line with the tradition of health informatics in which the authors work. For example, a scoping review by Daniore, Nittas, and von Wyl [71] found that in digital health studies of comparable duration, the target sample size was 72 participants (range 50–120 participants). While the current sample size allows us to draw meaningful inferences and conclusions, a larger sample would have been advantageous, for example, by enabling additional statistical analyses. On the other hand, narrative interview analysis of a smaller sample could help researchers develop a rich understanding of the concerns and misconceptions that may prevent individuals from obtaining needed medical help. 

It is also very important to conduct evaluation research around public education, outreach, and engagement programs. In order to develop effective interventions, we need to have extensive data on what works for different groups under different circumstances. Evaluation should focus on gauging the role of different agents and various cognitive and social factors in bringing about trust and opinion change.

Building trust that would enable professionals to help individuals understand and accept beneficial health practices, including vaccination, requires a sustained effort by researchers, professionals across a range of relevant fields (e.g., healthcare, education, and librarianship), and community leaders.

## Figures and Tables

**Figure 1 vaccines-11-00094-f001:**
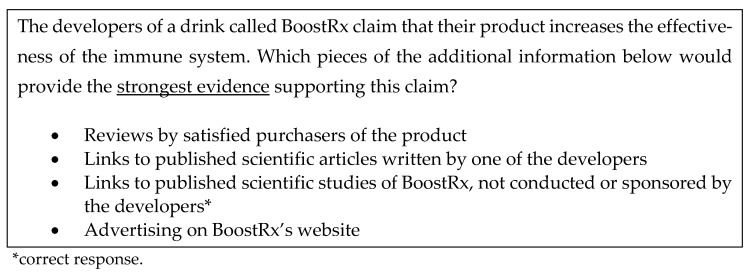
Sample information literacy question.

**Figure 2 vaccines-11-00094-f002:**
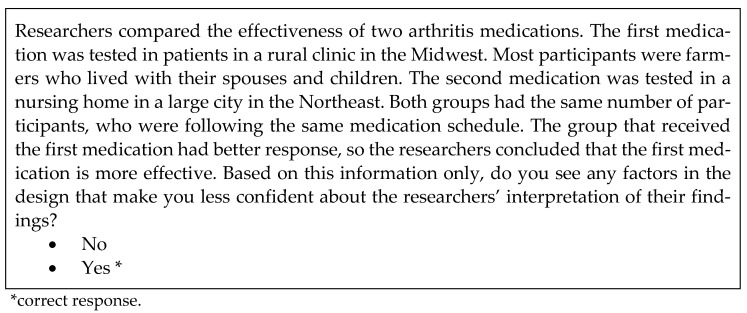
Sample science literacy question, non-equivalent samples.

**Figure 3 vaccines-11-00094-f003:**
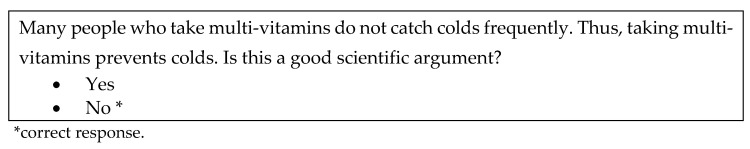
Sample science literacy question, self-report without control.

**Figure 4 vaccines-11-00094-f004:**
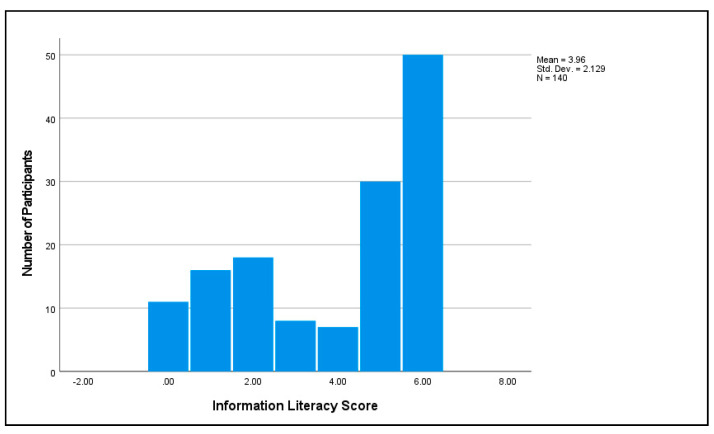
Participants’ information literacy.

**Figure 5 vaccines-11-00094-f005:**
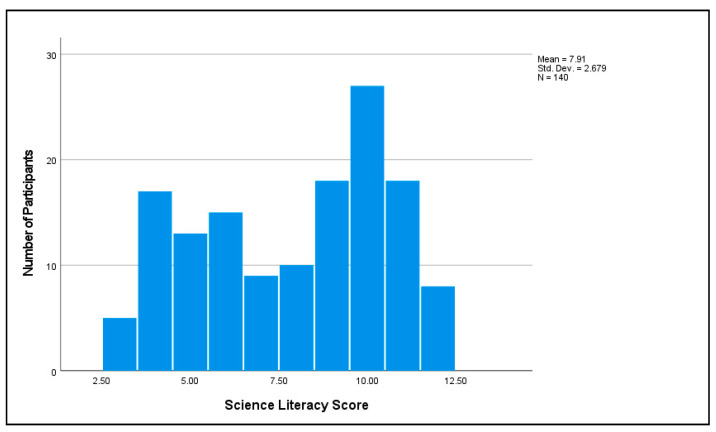
Participants’ science literacy.

**Table 1 vaccines-11-00094-t001:** Participants’ demographics.

Variable	Level	*n*	%
Age	18–29	31	22.14
	30–49	90	65.29
	50–64	17	12.14
	65+	2	1.43

Education	High school or less	17	12.14
	Some college	30	21.43
	College grad	73	52.14
	Postgraduate degree	20	14.29

Gender	Female	52	37.14
	Male	86	61.43
	Gender non-conforming, neither exclusively male nor female	1	0.71
	Decline to answer	1	0.71

Race	Asian	11	7.86
	White	102	72.86
	Black or African American	13	9.23
	Hispanic or Latino	8	5.71
	American Indian or Alaska Native	2	1.43
	Other	2	1.43
	Decline to answer	2	1.43

**Table 2 vaccines-11-00094-t002:** Participants’ religiosity, political ideology, and party affiliation.

Variable	Level	*n*	%
Importance of religion in life	Not at all important	50	35.71
	Slightly important	8	5.71
	Moderately important	21	15.00
	Very important	33	23.57
	Extremely important	28	20.00

Frequency of attending religious services	Seldom/never	68	48.57
	Once or twice a month/a few times a year	31	22.14
	At least once a week	41	29.29

Political party affiliation	Democrat or leaning Democrat	90	64.29
	No leaning/Independent	14	10.00
	Republican or leaning Republican	34	24.29
	Other	2	1.43

Political ideology	Very liberal	30	21.43
	Liberal	47	33.57
	Moderate	23	16.43
	Conservative	20	14.29
	Very conservative	19	13.57
	Not sure	1	0.71

**Table 3 vaccines-11-00094-t003:** Participants’ vaccination status in August 2021.

	Received at Least One Vaccine Dose	Planning to Get Vaccinated	Considering Vaccination	Not Interested in Vaccination
Number of participants	100	15	15	10
% participants	71.43 *	10.71	10.71	7.14

***** Compared to 64.5% among all Americans on 25 September 2021 [57].

**Table 4 vaccines-11-00094-t004:** Study variables.

Dependent Variables	Independent Variables
General attitude towards vaccination (General Vaccination Attitude) Attitude towards COVID-19 vaccines (Covid Vaccines Attitudes) Vaccination Status	Importance of religion in life (Importance of Religion) Frequency of attending religious services (Religious Attendance) Political party affiliation (Party Affiliation) Political ideology Information literacy Science literacy Attitudes towards science Interpersonal trust Public health trust

**Table 5 vaccines-11-00094-t005:** Single predictor models analyses.

Variable	General Vaccination Attitude			Covid Vaccines Attitudes			Vaccination Dtatus		
	F	*p*	Var *	F	*p*	Var *	F	*p*	Var *
Importance of religion	24.22 **	<0.001	15%	2.57	0.111	NA	1.52	0.212	NA
Religious attendance	13.37 **	<0.001	9%	0.6	0.808	NA	0.213	0.645	NA
Party affiliation	5.30 **	0.023	4%	12.76 **	<0.001	9%	6.29	0.13	NA
Political ideology	11.06 **	0.001	7%	8.90 **	0.003	6%	2.54	0.114	NA
Information literacy	39.09 **	<0.001	22%	3.11	0.08	NA	2.55	0.112	NA
Science literacy	24.50 **	<0.001	15%	0.044	0.833	NA	0.004	0.948	NA
General trust	2.72	0.101	NA	23.61 **	<0.001	15%	9.60 **	0.002	7%
Public health trust	62.87 **	<0.001	31%	145.27 **	<0.001	51%	45.72 **	<0.001	25%
Attitudes towards science	73.39 **	<0.001	35%	20.18 **	<0.001	13%	8.42 **	0.004	6%

* variance explained; ** statistically significant at *p* < 0.006 two-tailed Bonferroni-corrected for multiple hypotheses.

**Table 6 vaccines-11-00094-t006:** Effect-size ranked significant single predictors (%variance explained; Pearson’s r).

General Vaccination Attitudes	Covid Vaccines Attitudes	Vaccination Status
LARGE EFFECTS -attitudes towards science (35; 0.59) -public health trust (31; 0.56) MEDIUM EFFECTS -information literacy (22; 0.47) -science literacy (15; 0.39) -importance of religion (15; 0.39) -religious attendance (9; 0.30) SMALL EFFECTS -political ideology (7; 0.26) -party affiliation (4; 0.20)	LARGE EFFECT -public health trust (51; 0.71) MEDIUM EFFECTS -general trust (15; 0.39) -attitudes towards science (13; 0.36) -party affiliation (9; 0.30) SMALL EFFECT -political ideology (7; 0.26)	LARGE EFFECT -public health trust (25; 0.50) SMALL EFFECTS -general trust (7; 0.26) -attitudes towards science (6; 0.24)

SMALL: 0.1 < es ≤ 0.3; MEDIUM: 0.3 < es ≤ 0.5; LARGE: es > 0.5 [65].

**Table 7 vaccines-11-00094-t007:** Significant predictors of general and covid vaccination attitudes, multiple predictor model.

	General Vaccination Attitude		Covid Vaccination Attitude	
Predictor	t (1,139)	*p*	t (1139)	*p*
Attitudes towards science	3.71	<0.001	0.29 *	0.77 *
Public health trust	2.68	0.008	8.06	<0.001
General trust	1.90	0.059	2.09	0.038
Party affiliation	0.72 *	0.48 *	1.99	0.049

* not statistically significant.

**Table 8 vaccines-11-00094-t008:** Trust in public health authority institutions.

Authority	1—Do Not Trust at All	2	3	4	5—Trust Completely
Centers for Disease Control and Prevention, CDC	8	8	29	45	50
National Institutes of Health, NIH	8	5	28	53	46
Your primary doctor or healthcare provider	8	6	26	60	40
A major university that conducts biomedical research	4	7	65	60	34
A national health association, such as American Diabetes Association	5	10	37	57	31
Food and Drug Administration, FDA	6	8	39	62	25

**Table 9 vaccines-11-00094-t009:** The relationship between attitudes towards science and ideology.

Attitudes towards Science	Negative	Cautious	Positive
Liberal or very liberal (%)	5 (6%)	19 (25%)	53 (69%)
Moderate (%)	0 (0%)	8 (35%)	15 (65%)
Conservative or very conservative (%)	2 (5%)	24 (62%)	13 (33%)

**Table 10 vaccines-11-00094-t010:** Concerns about COVID-19 vaccines being unsafe or ineffective.

Concern	*n*	%
Vaccines not safe	37	26.43
Vaccines not effective	25	17.86
Already had COVID-19	17	12.14
Not afraid of COVID-19	15	10.71
Vaccination logistics inconvenient	8	5.71
Vaccination against religion	9	6.43
Other	6	4.29

**Table 11 vaccines-11-00094-t011:** Information sources.

Information Source	Participants	%
Newspapers	90	64.29
The CDC	85	60.71
Social media	83	59.29
The TV	76	54.29
Health and wellness publications and websites	64	45.71
Your primary doctor or healthcare provider	56	40.00
Sources created by your local governments	44	31.43
Original scientific articles	41	29.29
Web or mobile app forums (e.g., NextDoor)	28	20.00
People you know	25	17.86
Other	4	2.86

**Table 12 vaccines-11-00094-t012:** Trusted sources.

Information Source	Average Trusts Score
Your primary doctor or healthcare provider	4.09
Public Health Agencies	4.02
Librarians in a nearby hospital or medical school library	3.28
Friends and family	3.27
Librarians in your local public library	2.96
Journalists	2.91

## Data Availability

The data presented in this study are available on request from the corresponding author. The data are not publicly available due to privacy restrictions.

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
