# Peer review of "Cognitive and Cultural Factors That Affect General Vaccination and COVID-19 Vaccination Attitudes"

_vaccines, 2022, doi:10.3390/vaccines11010094_

Round 1

Reviewer 1 Report

Estimated Authors,

I've read with great interest your paper entitled: "Cognitive and Cultural Factors that Affect General Vaccination and Covid-19 Vaccination Attitudes".

This is a very small cross-section study (participants: 140) dealing with positive and negative effectors on vaccination (and particularly for COVID-19 vaccination). Briefly, Authors have identified a series of factors, including attitude towards sciences, and public health trust, literacy, religiosity, and ideology (including political ideology).

Despite its potential interest, the present paper cannot be accepted on Vaccines in its current stage of development for several reasons, and namely:

1) first and foremost, while the third year of the pandemic is slowly closing down, a great amount of evidence on SARS-CoV-2 vaccination has been collected; a small study can contribute to our general understanding of this main topic if developed in order to be innovative in a certain way. Authors have inquired study participants through a very broad questionnaire, that has assessed a lot of different aspects. Unfortunately, this great amount of information is reported in a non consistent way, through a great number of tables. I'm wondering whether the Authors could collapse some of the available Tables (e.g. 7-8, 11-12) in order to make the reported information more consistently included across the main text.

2) Authors should discuss and explain in a far deeper way why they opted in for recruiting participants through Amazon platform, which pros and cons has this platform compared to other ones. Moreover, unless Authors clearly state that the present one represent a convenience sample, they should explain whether a preventive sample size was performed or not, and way they did fail to achieve the minimum figures requires.

3) introduciton and methods sections are too long, and should be shortened.

4) The chapter on Information Literay should explain in further and more transparent way the menining of Figure 2/4 and how these information were actually integrated with the main text. If the aforementioned figures only represent a sample and an example of the items that were actually assessed, Authors should incude the correponding material by means of supplementary files. 

Author Response

Reviewer 1 Comment:

Collapse some of the available Tables (e.g. 7-8, 11-12) in order to make the reported information more consistently included across the main text.

Response:

As suggested, we merged Tables 7 and 8. As tables 11 and 12 presented different types of data – people’s concerns about Covid-18 vaccines (Table 11) and sources of Covid-19 information (Table 12), we felt it was not feasible to merge these two. However, we removed Table 14, collapsing its data into brief textual presentation. In the resulting document, all but one of the Results subsections have only one data table.

Reviewer 1 Comment:

Authors should discuss and explain in a far deeper way why they opted in for recruiting participants through Amazon platform, which pros and cons has this platform compared to other ones.

Response:

The following was added to the Participants section: “MTurk was chosen because it has a uniquely large U.S. worker base, estimated to be 75% of its worker membership (Difallah, Filatova, & Ipeirotis, 2018), which is our target group. Moreover, MTurk platform is well-known in the research community, making this sampling approach transparent to readers, and the authors have ample experience using the platform.”

The added reference is:

Difallah, D., Filatova,E., & Ipeirotis, P. (2018). Demographics and dynamics of Mechanical Turk workers. Proceedings of the Eleventh ACM International Conference on Web Search and Data Mining (WSDM '18). Association for Computing Machinery, New York, NY, USA, 135–143.

Reviewer 1 Comment:

Unless Authors clearly state that the present one represent a convenience sample, they should explain whether a preventive sample size was performed or not, and way they did fail to achieve the minimum figures requires.

Response:

We edited the Participants section to clarify that “A convenience sample of 140 participants were recruited via the Amazon Mechanical Turk (MTurk) crowdsourcing platform.” (The word “convenience” is not highlighted in the manuscript). We also added the following, “The sample size was determined by power assessment of our previous studies of health information behavior with a comparable number of variables (Keselman et al., 2021), as well as by time and budget constraints.”

The reference is:

Keselman, A., Arnott Smith, C., Leroy, G., & Kaufman, D. R. (2021). Factors influencing willingness to share health misinformation videos on the Internet: Web-based survey. Journal of Medical Internet Research23(12), e30323.

Reviewer 1 Comment:

Introduction and methods sections are too long, and should be shortened.

Response:

We tightened the Introduction content of the previous version, including the section on the Factors that Affect Vaccination; Science Literacy; Information Literacy; and Trust, Ideology, and Culture. Although content was also added to the Introduction in response to another reviewer’s queries (to address request for coverage of background additional literature), the total section word count was reduced from 2,331 to 1,892.

In the Methods section, we significantly reduced the Participant Demographics table and shortened Participants, Information Literacy, Science Literacy, and Public Health Trust sections. Some content was added to the Participants section to explain MTurk methodology in greater detail, as requested.

Reviewer 1 Comment:

The chapter on Information Literacy should explain in further and more transparent way the menining of Figure 2/4 and how these information were actually integrated with the main text. If the aforementioned figures only represent a sample and an example of the items that were actually assessed, Authors should incude the correponding material by means of supplementary files. 

Response:

We further clarified that the figures present sample test items and provided the full questionnaire as a supplementary file.

Reviewer 2 Report

The paper is conceptually weak. The empirical method follows the conceptualization. While the method itself is of acceptable quality, the conceptual flaws undermine the results and the conclusion. You may want to reframe the empirical work with an appropriate conceptual framework.

The title suggests the study is about cognitive and cultural factors that affect general vaccination and Covid-19 vaccination attitudes. The list of factors studied are information literacy, science literacy, attitudes towards science, interpersonal trust, public health trust, political ideology, and religiosity. I can guess some, but please clarify which of them are cognitive and which are cultural factors. The confusion is compounded by the statement: “In this study, we approach vaccination from a cognitive-emotional perspective, focusing on individual factors that influence beliefs about safety and effectiveness of Covid-9 vaccines and good will and competency of the establishment that endorses them.” I am not sure of the cognitive-cultural-emotional link. There is further a section on “Trust, Ideology, and Culture.” Is culture considered in conjunction with trust and ideology? Last, the specific objectives are described as follows: “Specific objectives of this study include investigating the impact of information literacy, science literacy, attitudes towards science, interpersonal trust, public health trust, political ideology and affiliation, and religiosity on three dependent variables: 1) general attitude towards vaccination, 2) attitude towards Covid-19 vaccination, and 3) Covid-19 vaccination status. We also aimed to characterize the distribution of key independent variables in the sample.” Notice the introduction of a third dependent variable. And what is it about the distribution of key independent variables?

To this reviewer the conceptual framework is neither logical nor coherent. It appears to be a framework of convenience and not based on a comprehensive understanding and analysis of the problem. Implicitly, it appears to be a US-based framework and not a universal scientific framework. I would like to be convinced otherwise.

Author Response

Reviewer 2 Comment:

The paper is conceptually weak. The empirical method follows the conceptualization. While the method itself is of acceptable quality, the conceptual flaws undermine the results and the conclusion. You may want to reframe the empirical work with an appropriate conceptual framework.

The title suggests the study is about cognitive and cultural factors that affect general vaccination and Covid-19 vaccination attitudes. The list of factors studied are information literacy, science literacy, attitudes towards science, interpersonal trust, public health trust, political ideology, and religiosity. I can guess some, but please clarify which of them are cognitive and which are cultural factors. The confusion is compounded by the statement: “In this study, we approach vaccination from a cognitive-emotional perspective, focusing on individual factors that influence beliefs about safety and effectiveness of Covid-9 vaccines and good will and competency of the establishment that endorses them.” I am not sure of the cognitive-cultural-emotional link. There is further a section on “Trust, Ideology, and Culture.” Is culture considered in conjunction with trust and ideology? Last, the specific objectives are described as follows: “Specific objectives of this study include investigating the impact of information literacy, science literacy, attitudes towards science, interpersonal trust, public health trust, political ideology and affiliation, and religiosity on three dependent variables: 1) general attitude towards vaccination, 2) attitude towards Covid-19 vaccination, and 3) Covid-19 vaccination status. We also aimed to characterize the distribution of key independent variables in the sample.” Notice the introduction of a third dependent variable.

To this reviewer the conceptual framework is neither logical nor coherent. It appears to be a framework of convenience and not based on a comprehensive understanding and analysis of the problem. Implicitly, it appears to be a US-based framework and not a universal scientific framework. I would like to be convinced otherwise.

Response:

We appreciate this critique, which led us to edit the introduction and discussion section significantly, highlighting theoretical underpinnings of our work and linking it to empirical findings of other authors.

The following was added to the introduction:

“In this study, we approach vaccination from a cognitive-cultural perspective, focusing on individual factors. Cognitive studies explain that individuals selectively process information, giving more weight to arguments that are coextensive with their biases (Modgil et al., 2921). These cognitive factors impact on opinion-formation and decision-making, and interplay with social and cultural factors. Individuals become trapped in homogenous information echo chambers with those who have similar interests and beliefs (Dow et al., 2021). Kahan’s (2017) identity protective mechanism (IPC) construct further explains the individual tendency to accept or reject beliefs associated with particular groups, such as political parties. The IPC can also serve as a filter for information sources and institutions that are deemed trustworthy or not. Personal affiliations hold more sway, and may cause an individual to bypass rational thought, deliberative processes and factually accurate beliefs.

We investigate two cognitive factors, science literacy, including attitudes toward science, and information literacy. These factors are sets of skills and knowledge essential to navigating information and psychologically predisposing individuals to certain beliefs about vaccination safety and efficacy (Norman & Skinner, 2006). Cultural factors considered in this study pertain to political ideology / party affiliation and religiosity. We also focus on several kinds of trust, in which the cognitive and the sociocultural are closely intertwined.”

The addition aims to deliver the following point:

  • The factors our study investigated are most precisely defined as cognitive and cultural, even if vaccination is a charge issue where emotions are intertwined cultural beliefs and decision-making. To reflect this, we sharpened the language removing references to emotional factors.
  • The new iteration of the paper is explicit about our separation of the factors into cognitive (science literacy and information literacy) and cultural (those pertain to political ideology / party affiliation and religiosity). We also specify that cognition and culture cannot be separated when considering trust.
  • While our study is empirically-driven and the whole range of factors we consider are the ones that have appeared in general and Covid-19 vaccine hesitancy studies, our cognitive factors are tied to the Norman and Skinner 2006 e-Health literacy that comprises of six core or literacies: (1) traditional literacy, (2) health literacy, (3) information literacy, (4) scientific literacy, (5) media literacy, and (6) computer literacy. As the authors’ research program is grounded in the field of consumer health informatics, much of our work is driven by this framework. In the case of this paper, after some consideration, we decided to cite the paper but limit the mention of specific literacies to the ones directly relevant to the study. As this study is not specifically focused on e-health, we felt that describing the full framework will add gratuitous complexity for the readers. We also want to thank the reviewer for the suggestion to add references to other studies of cultural factors we explored. While those were selected based on their empirical relevance, explicitly linking our choices to recent literature about Covid-19 vaccination strengthens the paper.

The question of adding Covid-19 vaccination status as the third outcome variable we would like to address separately. Our primary theoretical interest is vaccination attitudes, rather than behavior, behavior is affected by many pragmatic factors such as availability of vaccine, hours vaccination sites are open, and transportation to vaccination sites. Thus, our title focuses on the factors of theoretical interest. At the same time, from the public health perspective, achieving vaccination is the ultimate goal of attitude change, so we felt it was important to include the variable in the analysis.

The reviewer writes “It appears to be a framework of convenience and not based on a comprehensive understanding and analysis of the problem.” As mentioned above, the framework is driven by a combination of the Norman and Skinner 2006 e-Health literacy model and empirical review. While our original manuscript did not adequately describe the empirical foundation, the revised version explicitly links the factors we investigate to other empirical studies. We thank this reviewer for the invaluable help in collecting research from a range of countries to highlight this empirical foundation.

Finally, with regard to the foundation for the study, this reviewer writes “Implicitly, it appears to be a US-based framework and not a universal scientific framework.” The study was conducted in the US and our initial literature review drew primarily on US-based studies. The edited version expands the literature review with many international studies from around the world (see our responses to Reviewer 3), showing how factors such as literacies, trust in authorities, and political attitudes affected Covid-19 vaccination openness and hesitancy around the globe. The revised introduction makes this comparison explicit. The following segment is an example: “Across the world, trust in the government, the healthcare system, and civil society correlates with Covid-19 vaccine acceptance and uptake (Chen, Lee, & Lin, 2022; Cvjetkovic et al., 2022, De Freitas, Basdeo, & Wang, 2021). In the US, studies reveal distrust pharmaceutical companies, being perceived as motivated by profit (Jamison et al., 2019) and, overall, trust in the U.S. federal government has been waning since the Vietnam War era (Cooper, 2018).” The following was also added: “While the study was conducted from the US perspective, literature suggests that similar factors are influential worldwide.”

Reviewer 3 Report

Cognitive and Cultural Factors that Affect General Vaccination and Covid-19 Vaccination Attitudes”(vaccines-2036592

This manuscript aims to determine the cognitive and cultural factors that affect the General Vaccination and Covid-19 Vaccination Attitudes. Specifically, information literacy, science literacy, attitudes towards science, interpersonal trust, public health trust, political ideology, and religiosity was investigated. The results revealed that General vaccination attitude was most affected by attitude towards science and public health trust, and to a lesser degree by information literacy, science literacy, and religiosity. Attitudes towards Covid-19 vaccines were most affected by public health trust, and to a lesser extent by general trust, ideology, and attitude towards science. Vaccination status was most influenced by public health trust. Overall, this topic is important and timely. However, some concerns appeared after reading the whole manuscript.

1. The first and most important concerns is the lack of appropriate literature review and the identification of research gaps, which makes the readers wonder the motivations to do the current study and the novelties of the current study. Some important and directly related are listed below.

Reviews:

Shakeel, C. S., Mujeeb, A. A., Mirza, M. S., Chaudhry, B., & Khan, S. J. (2022). Global COVID-19 vaccine acceptance: a systematic review of associated social and behavioral factors. Vaccines, 10(1), 110.

Pires, C. (2022). Global predictors of COVID-19 vaccine hesitancy: A systematic review. Vaccines, 10(8), 1349.

Sallam, M., Al-Sanafi, M., & Sallam, M. (2022). A global map of COVID-19 vaccine acceptance rates per country: an updated concise narrative review. Journal of Multidisciplinary Healthcare15, 21.

Dubé, È., Ward, J. K., Verger, P., & MacDonald, N. E. (2021). Vaccine hesitancy, acceptance, and anti-vaccination: trends and future prospects for public health. Annu Rev Public Health42(1), 175-91.

Barello, S., Acampora, M., Paleologo, M., Schiavone, L., Anderson, G., & Graffigna, G. (2022). Public views on the Covid19 immunity certificate: A scoping review. Health Expectations.

AlAmer, R., Maneze, D., Everett, B., Montayre, J., Villarosa, A. R., Dwekat, E., & Salamonson, Y. (2022). COVID19 vaccination intention in the first year of the pandemic: A systematic review. Journal of clinical nursing31(1-2), 62-86.

Majid, U., Ahmad, M., Zain, S., Akande, A., & Ikhlaq, F. (2022). COVID-19 vaccine hesitancy and acceptance: a comprehensive scoping review of global literature. Health promotion international37(3), daac078.

Literature about science literacy

Lindeman, M., Svedholm-Häkkinen, A. M., & Riekki, T. J. (2022). Searching for the cognitive basis of anti-vaccination attitudes. Thinking & Reasoning, 1-26.

Motoki, K., Saito, T., & Takano, Y. (2021). Scientific literacy linked to attitudes toward COVID-19 vaccinations: A pre-registered study.

Siani, A., Carter, I., & Moulton, F. (2022). Political views and science literacy as indicators of vaccine confidence and COVID-19 concern. Journal of Preventive Medicine and Hygiene63(2), E257.

Hu, M., Jia, H., & Xie, Y. (2021). Passport to a mighty nation: Exploring sociocultural foundation of Chinese public’s attitude to COVID-19 vaccine certificates. International Journal of Environmental Research and Public Health18(19), 10439.

Literature about information literacy

Hicks, A., & Lloyd, A. (2022). Agency and liminality during the COVID-19 pandemic: Why information literacy cannot fix vaccine hesitancy. Journal of Information Science, 01655515221124003.

Gu, C., & Feng, Y. (2022). Influence of Public Engagement with Science on Scientific Information Literacy During the COVID‑19 Pandemic. Science & education31(3), 619-633.

Xin, M., Luo, S., Wang, S., Zhao, J., Zhang, G., Li, L., ... & Lau, J. T. F. (2022). The Roles of Information Valence, Media Literacy and Perceived Information Quality on the Association Between Frequent Social Media Exposure and COVID-19 Vaccination Intention. American Journal of Health Promotion, 08901171221121292.

Kruijt, J., Meppelink, C. S., & Vandeberg, L. (2022). Stop and think! Exploring the role of news truth discernment, information literacy, and impulsivity in the effect of critical thinking recommendations on trust in fake COVID-19 news. European Journal of Health Communication3(2), 40-63.

Austin, E. W., Austin, B. W., Borah, P., Domgaard, S., & McPherson, S. M. (2022). How Media Literacy, Trust of Experts and Flu Vaccine Behaviors Associated with COVID-19 Vaccine Intentions. American Journal of Health Promotion, 08901171221132750.

Besides science literacy and information literacy,  literature about science literacy and COVID-19 vaccine literacy should also mentioned:

Okan, O., Messer, M., Levin-Zamir, D., Paakkari, L., & Sørensen, K. (2022). Health literacy as a social vaccine in the COVID-19 pandemic. Health Promotion International.

Bin Naeem, S., & Kamel Boulos, M. N. (2021). COVID-19 misinformation online and health literacy: a brief overview. International journal of environmental research and public health18(15), 8091.

Takahashi, Y., Ishitsuka, K., Sampei, M., Okawa, S., Hosokawa, Y., Ishiguro, A., ... & Morisaki, N. (2022). COVID-19 vaccine literacy and vaccine hesitancy among pregnant women and mothers of young children in Japan. Vaccine.

Literature about trust

De Freitas, L., Basdeo, D., & Wang, H. I. (2021). Public trust, information sources and vaccine willingness related to the COVID-19 pandemic in Trinidad and Tobago: an online cross-sectional survey. The Lancet Regional Health-Americas3, 100051.

Rodriguez-Morales, A. J., & Franco, O. H. (2021). Public trust, misinformation and COVID-19 vaccination willingness in Latin America and the Caribbean: today's key challenges. The Lancet Regional Health–Americas3.

Chen, X., Lee, W., & Lin, F. (2022). Infodemic, institutional trust, and COVID-19 vaccine hesitancy: A cross-national survey. International journal of environmental research and public health19(13), 8033.

Cvjetkovic, S., Jeremic Stojkovic, V., Mandic-Rajcevic, S., Matovic-Miljanovic, S., Jankovic, J., Jovic Vranes, A., ... & Stamenkovic, Z. (2022). Societal Trust Related to COVID-19 Vaccination: Evidence from Western Balkans. Sustainability14(20), 13547.

Literature about political ideology

Agarwal, R., Dugas, M., Ramaprasad, J., Luo, J., Li, G., & Gao, G. (2021). Socioeconomic privilege and political ideology are associated with racial disparity in COVID-19 vaccination. Proceedings of the National Academy of Sciences118(33), e2107873118.

Debus, M., & Tosun, J. (2021). Political ideology and vaccination willingness: Implications for policy design. Policy sciences54(3), 477-491.

Park, H. K., Ham, J. H., Jang, D. H., Lee, J. Y., & Jang, W. M. (2021). Political ideologies, government trust, and COVID-19 vaccine hesitancy in South Korea: a cross-sectional survey. International journal of environmental research and public health18(20), 10655.

Literature about religiosity

Garcia, L. L., & Yap, J. F. C. (2021). The role of religiosity in COVID-19 vaccine hesitancy. Journal of Public Health43(3), e529-e530.

Kesgin, M., Can, A. S., Gursoy, D., Ekinci, Y., & Aldawodi, K. (2022). Effects of religiosity and travel desire on COVID-19 vaccination intentions. Current Issues in Tourism, 1-17.

Orlandi, L. B., Febo, V., & Perdichizzi, S. (2022). The role of religiosity in product and technology acceptance: Evidence from COVID-19 vaccines. Technological Forecasting and Social Change185, 122032.

Milligan, M. A., Hoyt, D. L., Gold, A. K., Hiserodt, M., & Otto, M. W. (2022). COVID-19 vaccine acceptance: Influential roles of political party and religiosity. Psychology, Health & Medicine27(9), 1907-1917.

López-Cepero, A., Rodríguez, M., Joseph, V., Suglia, S. F., Colón-López, V., Toro-Garay, Y. G., ... & Pérez, C. M. (2022). Religiosity and Beliefs toward COVID-19 Vaccination among Adults in Puerto Rico. International Journal of Environmental Research and Public Health19(18), 11729.

Other related references:

Kricorian, K., Civen, R., & Equils, O. (2022). COVID-19 vaccine hesitancy: Misinformation and perceptions of vaccine safety. Human Vaccines & Immunotherapeutics18(1), 1950504.

Arbel, Y., Arbel, Y., Kerner, A., & Kerner, M. (2022). Covid 19 vaccination: Accessibility or literacy? Israel as a case study. International Journal of Disaster Risk Reduction71, 102794.

Engelbrecht, M. C., Kigozi, N. G., & Heunis, J. C. (2022). Factors Associated with Limited Vaccine Literacy: Lessons Learnt from COVID-19. Vaccines10(6), 865.

2. Because the lack of the appropriate literature review, then the current findings were not well discussed with previous findings.

3. How did you determine the sample size? Did you calculate the sample size needed before formal study?

References:

Lakens, D. (2022). Sample size justification. Collabra: Psychology8(1), 33267.

Although you mentioned this limitation in the current manuscript, however, N=140 is too little to get reliable results, especially when considering that you included so many variables in your study.

4. The presentation of results could be more concise.

5. Conclusions should be Discussionin the current manuscript. And a concise Conclusions part should be provided at the end of this manuscript.

6. I recommend that the paper be thoroughly proofread and edited for languages and grammars, to enhance readership.

Author Response

Reviewer 3 Comment:

The reviewer recommended citing from the following literature reviews:

  1. Shakeel, C. S., Mujeeb, A. A., Mirza, M. S., Chaudhry, B., & Khan, S. J. (2022). Global COVID-19 vaccine acceptance: a systematic review of associated social and behavioral factors. Vaccines, 10(1), 110.
  2. Pires, C. (2022). Global predictors of COVID-19 vaccine hesitancy: A systematic review. Vaccines, 10(8), 1349.
  3. Sallam, M., Al-Sanafi, M., & Sallam, M. (2022). A global map of COVID-19 vaccine acceptance rates per country: an updated concise narrative review. Journal of Multidisciplinary Healthcare, 15, 21.
  4. Dubé, È., Ward, J. K., Verger, P., & MacDonald, N. E. (2021). Vaccine hesitancy, acceptance, and anti-vaccination: trends and future prospects for public health. Annu Rev Public Health, 42(1), 175-91.
  5. Barello, S., Acampora, M., Paleologo, M., Schiavone, L., Anderson, G., & Graffigna, G. (2022). Public views on the Covid‐19 immunity certificate: A scoping review. Health Expectations.
  6. Al‐Amer, R., Maneze, D., Everett, B., Montayre, J., Villarosa, A. R., Dwekat, E., & Salamonson, Y. (2022). COVID‐19 vaccination intention in the first year of the pandemic: A systematic review. Journal of clinical nursing, 31(1-2), 62-86.
  7. Majid, U., Ahmad, M., Zain, S., Akande, A., & Ikhlaq, F. (2022). COVID-19 vaccine hesitancy and acceptance: a comprehensive scoping review of global literature. Health promotion international, 37(3), daac078.

Response:

We reviewed the suggested papers, incorporating and citing references # 1,2,3,6,7

Reviewer 3 Comment:

The reviewer recommended citing from the following list of science literacy studies:

  1. Lindeman, M., Svedholm-Häkkinen, A. M., & Riekki, T. J. (2022). Searching for the cognitive basis of anti-vaccination attitudes. Thinking & Reasoning, 1-26.
  2. Motoki, K., Saito, T., & Takano, Y. (2021). Scientific literacy linked to attitudes toward COVID-19 vaccinations: A pre-registered study.
  3. Siani, A., Carter, I., & Moulton, F. (2022). Political views and science literacy as indicators of vaccine confidence and COVID-19 concern. Journal of Preventive Medicine and Hygiene63(2), E257.
  4. Hu, M., Jia, H., & Xie, Y. (2021). Passport to a mighty nation: Exploring sociocultural foundation of Chinese public’s attitude to COVID-19 vaccine certificates. International Journal of Environmental Research and Public Health18(19), 10439.

Response:

We reviewed the suggested papers, incorporating and citing references # 1,2,3

Reviewer 3 Comment:

The reviewer recommended citing from the following list of information literacy studies:

  1. Hicks, A., & Lloyd, A. (2022). Agency and liminality during the COVID-19 pandemic: Why information literacy cannot fix vaccine hesitancy. Journal of Information Science, 01655515221124003.
  2. Gu, C., & Feng, Y. (2022). Influence of Public Engagement with Science on Scientific Information Literacy During the COVID‑19 Pandemic. Science & education31(3), 619-633.
  3. Xin, M., Luo, S., Wang, S., Zhao, J., Zhang, G., Li, L., ... & Lau, J. T. F. (2022). The Roles of Information Valence, Media Literacy and Perceived Information Quality on the Association Between Frequent Social Media Exposure and COVID-19 Vaccination Intention. American Journal of Health Promotion, 08901171221121292.
  4. Kruijt, J., Meppelink, C. S., & Vandeberg, L. (2022). Stop and think! Exploring the role of news truth discernment, information literacy, and impulsivity in the effect of critical thinking recommendations on trust in fake COVID-19 news. European Journal of Health Communication3(2), 40-63.
  5. Austin, E. W., Austin, B. W., Borah, P., Domgaard, S., & McPherson, S. M. (2022). How Media Literacy, Trust of Experts and Flu Vaccine Behaviors Associated with COVID-19 Vaccine Intentions. American Journal of Health Promotion, 08901171221132750.

Response:

We reviewed the suggested papers, incorporating and citing references # 3 and 4

Reviewer 3 Comment:

The reviewer recommended citing from this additional list of studies:

Besides science literacy and information literacy,  literature about science literacy and COVID-19 vaccine literacy should also mentioned:

  1. Okan, O., Messer, M., Levin-Zamir, D., Paakkari, L., & Sørensen, K. (2022). Health literacy as a social vaccine in the COVID-19 pandemic. Health Promotion International.
  2. Bin Naeem, S., & Kamel Boulos, M. N. (2021). COVID-19 misinformation online and health literacy: a brief overview. International journal of environmental research and public health18(15), 8091.
  3. Takahashi, Y., Ishitsuka, K., Sampei, M., Okawa, S., Hosokawa, Y., Ishiguro, A., ... & Morisaki, N. (2022). COVID-19 vaccine literacy and vaccine hesitancy among pregnant women and mothers of young children in Japan. Vaccine.

Response:

We reviewed the suggested papers, incorporating and citing all of them

Reviewer 3 Comment:

The reviewer recommended citing from the following trust literature:

  1. De Freitas, L., Basdeo, D., & Wang, H. I. (2021). Public trust, information sources and vaccine willingness related to the COVID-19 pandemic in Trinidad and Tobago: an online cross-sectional survey. The Lancet Regional Health-Americas3, 100051.
  2. Rodriguez-Morales, A. J., & Franco, O. H. (2021). Public trust, misinformation and COVID-19 vaccination willingness in Latin America and the Caribbean: today's key challenges. The Lancet Regional Health–Americas3.
  3. Chen, X., Lee, W., & Lin, F. (2022). Infodemic, institutional trust, and COVID-19 vaccine hesitancy: A cross-national survey. International journal of environmental research and public health19(13), 8033.
  4. Cvjetkovic, S., Jeremic Stojkovic, V., Mandic-Rajcevic, S., Matovic-Miljanovic, S., Jankovic, J., Jovic Vranes, A., ... & Stamenkovic, Z. (2022). Societal Trust Related to COVID-19 Vaccination: Evidence from Western Balkans. Sustainability14(20), 13547.

Response:

We reviewed the suggested papers, incorporating and citing references # 1,3, and 4

Reviewer 3 Comment:

The reviewer recommended citing from the following literature about political ideology:

  1. Agarwal, R., Dugas, M., Ramaprasad, J., Luo, J., Li, G., & Gao, G. (2021). Socioeconomic privilege and political ideology are associated with racial disparity in COVID-19 vaccination. Proceedings of the National Academy of Sciences118(33), e2107873118.
  2. Debus, M., & Tosun, J. (2021). Political ideology and vaccination willingness: Implications for policy design. Policy sciences54(3), 477-491.
  3. Park, H. K., Ham, J. H., Jang, D. H., Lee, J. Y., & Jang, W. M. (2021). Political ideologies, government trust, and COVID-19 vaccine hesitancy in South Korea: a cross-sectional survey. International journal of environmental research and public health18(20), 10655.

Response:

We reviewed the suggested papers, incorporating and citing references # 1 and 2

Reviewer 3 Comment:

The reviewer recommended citing from the following literature about political ideology:

  1. Garcia, L. L., & Yap, J. F. C. (2021). The role of religiosity in COVID-19 vaccine hesitancy. Journal of Public Health43(3), e529-e530.
  2. Kesgin, M., Can, A. S., Gursoy, D., Ekinci, Y., & Aldawodi, K. (2022). Effects of religiosity and travel desire on COVID-19 vaccination intentions. Current Issues in Tourism, 1-17.
  3. Orlandi, L. B., Febo, V., & Perdichizzi, S. (2022). The role of religiosity in product and technology acceptance: Evidence from COVID-19 vaccines. Technological Forecasting and Social Change185, 122032.
  4. Milligan, M. A., Hoyt, D. L., Gold, A. K., Hiserodt, M., & Otto, M. W. (2022). COVID-19 vaccine acceptance: Influential roles of political party and religiosity. Psychology, Health & Medicine27(9), 1907-1917.
  5. López-Cepero, A., Rodríguez, M., Joseph, V., Suglia, S. F., Colón-López, V., Toro-Garay, Y. G., ... & Pérez, C. M. (2022). Religiosity and Beliefs toward COVID-19 Vaccination among Adults in Puerto Rico. International Journal of Environmental Research and Public Health19(18), 11729.

Response:

We reviewed the suggested papers, incorporating and citing references # 3 and 4.

Reviewer 3 Comment:

The reviewer recommended considering the following additional references:

  1. Kricorian, K., Civen, R., & Equils, O. (2022). COVID-19 vaccine hesitancy: Misinformation and perceptions of vaccine safety. Human Vaccines & Immunotherapeutics18(1), 1950504.
  2. Arbel, Y., Arbel, Y., Kerner, A., & Kerner, M. (2022). Covid 19 vaccination: Accessibility or literacy? Israel as a case study. International Journal of Disaster Risk Reduction71, 102794.
  3. Engelbrecht, M. C., Kigozi, N. G., & Heunis, J. C. (2022). Factors Associated with Limited Vaccine Literacy: Lessons Learnt from COVID-19. Vaccines10(6), 865.

Response:

We reviewed the suggested papers, incorporating and citing references # 2 and 3.

Reviewer 3 Comment:

Because the lack of the appropriate literature review, then the current findings were not well discussed with previous findings.

Response:

In the revised version, we drew on the findings of the cited research, showing consistencies. For example, the following text was added to the discussion: “ Several factors emerging as relevant to vaccination attitudes in our study converge with other investigations of predictors of Covid-19 vaccination attitudes. For example, our finding of the significance of public health trust is consistent with Chen, Lee, and Lin (2022), Cvjetkovic et al., (2022), and De Freitas, Basdeo, and Wang (2021) findings about the positive impact of trust in the government, healthcare, and civil society on openness to Covid-19 vaccination. The importance of party affiliation is consistent with Khubchandani et al., 2021 and Agarwal et al. (2021), and the importance of information literacy with Engelbrecht, Kigozi, and Heunis (2022) and Takahashi et al. (2022). Religiosity, which has been shown to affect Covid-19 vaccination attitudes (Milligan et al., 2022; Orlandi, Febo and Perdichizzi, 2022), was relevant to general, but not Covid-19 specific, vaccination attitudes in our study.”

Reviewer 3 Comment:

How did you determine the sample size? Did you calculate the sample size needed before formal study? … Although you mentioned this limitation in the current manuscript, however, N=140 is too little to get reliable results, especially when considering that you included so many variables in your study.

Response:

The following explanation was added: “The sample size was determined by power assessment of our previous studies of health information behavior with a comparable number of variables (Keselman et al., 2021), as well as by time and budget constraints.”

The added reference is:

Keselman, A., Arnott Smith, C., Leroy, G., & Kaufman, D. R. (2021). Factors influencing willingness to share health misinformation videos on the Internet: Web-based survey. Journal of Medical Internet Research23(12), e30323.

 Reviewer 3 Comment:

The presentation of results could be more concise.

Response:

To make the presentation more concise, we merged Tables 7 and 8 and removed Table 14, collapsing its data into brief textual presentation.

Reviewer 3 Comment:

“Conclusions” should be “Discussion” in the current manuscript. And a concise “Conclusions” part should be provided at the end of this manuscript.

Response:

In order not to add to the length of the paper, we renamed “Conclusions” into “Discussion and Conclusions.”

Reviewer 3 Comment:

I recommend that the paper be thoroughly proofread and edited for languages and grammars, to enhance readership.

Response:

The edited draft was subjected to the slow read by all authors, with the final read by Catherine Arnott Smith who used to be an English editor before switching her career to health informatics research.

Round 2

Reviewer 1 Report

Estimated Authors,

I've appreciated the considerable efforts you paid in order to cope with my previous recommendations.

I've no further requests; please only be cautious when dealing with p value notations, as it is quite inconsistent across the main text (sometimes figures are reported with an initial zero (e.g. 0.001) sometimes without (e.g. .001).

I'm endorsing the eventual acceptance of the paper.

Reviewer 2 Report

The authors have responded adequately to my earlier comments.

Reviewer 3 Report

Thanks for the revisions. After revisions, the literature review have improved and the presentation of results became clearer. Howeve, some core concerns still remains. 

1. The determination of sample size is not satisfactory. Based on the author’s response, the current sample size was similar to their previous study.

Keselman, A., Arnott Smith, C., Leroy, G., & Kaufman, D. R. (2021). Factors influencing willingness to share health misinformation videos on the Internet: Web-based survey. Journal of Medical Internet Research23(12), e30323.

However, this number of sample size of this previous study is not based on a scientific evidence as well. Thus, the results retrieved from this small sample size might not not reliable.

2.  Besides the obejectives, the reseach gaps in current literature should also be clearly stated and how this investigation resolved these gaps?

3. It seems that only consistencies was revealed between the current findings and previous results. The discussion about the discrepancies might be more important and provide more insights for the future studies. 

Author Response

Reviewer 3 Comment:

The determination of sample size is not satisfactory. Based on the author’s response, the current sample size was similar to their previous study.

Keselman, A., Arnott Smith, C., Leroy, G., & Kaufman, D. R. (2021). Factors influencing willingness to share health misinformation videos on the Internet: Web-based survey. Journal of Medical Internet Research23(12), e30323.

However, this number of sample size of this previous study is not based on a scientific evidence as well. Thus, the results retrieved from this small sample size might not not reliable.

Response:

The following was added to the discussion section as a response:

The sample size employed in this study is based on heuristics, precedent, and pragmatic considerations. This choice was made because it would have been impossible to do a power calculation given the relatively scant literature on the investigated issues and the absence of effect-size measures. Our sample size is in line with the tradition of health informatics in which the authors work.  For example, a scoping review by Daniore, Nittas, and von Wyl (2022) found that in digital health studies of comparable duration, target sample size was 72 participants (range 50-120 participants). While the current sample size allows us to draw meaningful inferences and conclusions, a larger sample would have been advantageous, for example, enabling additional statistical analyses.

Reviewer 3 Comment:

Besides the objectives, the research gaps in current literature should also be clearly stated and how this investigation resolved these gaps?

Response:

The following was added to the introduction:

Calleja et al. (2021) proposed a public health research agenda emanating from the first WHO Infodemiology Conference. These authors note the paucity of research into the relationship between offline and online behaviors and suggest that more work is needed to understand “indicators that predict behaviors or serve as proxies for specific behaviors.”  The present work attempts to fill these gaps, first, by exploring the connections between vaccination attitudes and multiple cognitive and cultural factors, and second, by uncovering the differences between general vaccination attitudes and COVID-19 specific attitudes in respect to information behaviors. The cognitive and cultural factors are not usually considered in combination, and some of them (e.g., science literacy defined as knowledge about science) have not previously been considered in studies of vaccination hesitancy.

Reviewer 3 Comment:

 It seems that only consistencies was revealed between the current findings and previous results. The discussion about the discrepancies might be more important and provide more insights for the future studies. 

Response:

The following was added to the discussion:

Although our finding of the lack of impact of science literacy on Covid-19 vaccination attitude is discrepant with Siani, Carter, and Moulton (2022) and Motoki, Saito, and Takano (2021), those studies conceptualized science literacy as science education of knowledge, rather than knowledge about science.

And

Present finding of the lack of association between religiosity and attitudes toward COVID-19 vaccination constitutes an intriguing discrepancy from findings by Milligan et al. (2022) and Orlandi, Febo and Perdichizzi (2022).

Round 3

Reviewer 3 Report

Thanks for the revisions. Only one minor concern remains.

“NS” in Table 7 should be replaced by specfic data.

Author Response

Thank you very much for your help in improving the paper. NS in Table 7 was replaced with values.